# New Challenges Facing Systemic Therapies of Advanced HCC in the Era of Different First-Line Immunotherapy-Based Combinations

**DOI:** 10.3390/cancers14235868

**Published:** 2022-11-28

**Authors:** Julien Edeline, Tim Meyer, Jean-Frédéric Blanc, Jean-Luc Raoul

**Affiliations:** 1INSERM, Department of Medical Oncology, University Rennes, CLCC Eugène Marquis, COSS (Chemistry Oncogenesis Stress Signaling)—UMR_S 1242, F-35000 Rennes, France; 2Medical Oncology, University College London, London WC1E 6BT, UK; 3Hepatology, CHU Bordeaux, 33076 Bordeaux, France; 4Medical Oncology, Institut de Cancérologie de l’Ouest, 44805 Nantes, France

**Keywords:** hepatocellular carcinoma, systemic therapies, immunotherapy, anti-PD-1

## Abstract

**Simple Summary:**

In this review, we detail the data on a new treatment of hepatocellular carcinoma. A new combination of therapies, based on immunotherapy, has recently emerged as efficacious treatment. While being very interesting for patients, this raises new questions: how should we select the most appropriate treatment for each patient? What treatment can we offer after failure of these new treatments? How can we develop new combinations?

**Abstract:**

The standard of care of first-line systemic therapy for advanced hepatocellular carcinoma (HCC) is currently changing with the results of the IMbrave150 trial which are demonstrating superiority of the atezolizumab-bevacizumab combination over sorafenib, modifying this line of treatment for the first time in over 10 years. Recently, other immunotherapy-based combinations (durvalumab-tremelimumab, lenvatinib-pembrolizumab, cabozantinib-atezolizumab, and camrelizumab-rivoceranib) reported results in phase III studies, and might challenge this new standard of care. This revolution will lead to a considerable change in practice, and highlight challenges for future drug development. In this review, we will, firstly, describe results of the different combinations, and discuss the difficulties in selecting the first-line treatment. We will then present the different recommendations about second-line treatment following the first-line immunotherapy-based combination, discussing the rationale for the differences in existing recommendations. We will finally discuss the challenges for future drug development in advanced HCC.

## 1. Introduction

Hepatocellular carcinoma (HCC) is a leading cause of cancer death worldwide [1]. Main risk factors include hepatitis B and C virus infection, alcohol consumption, and, increasingly, non-alcoholic fatty liver disease (NAFLD). The relative role of these risks factors varies between countries and with time, with HBV being more prominent in China, HCV being more prominent in Japan, and alcohol being more prominent in some Western countries, while NAFLD is rising worldwide [2]. The Barcelona Clinic of Liver Cancer (BCLC) classification is the main algorithm used in Western guidelines [3]. Systemic treatment is currently used mainly in BCLC C patients, corresponding to advanced disease with either extrahepatic spread, macrovascular invasion, or in BCLC B patients with disease that is unsuitable for, or refractory to, locoregional treatments. Cytotoxic chemotherapy has not been clearly demonstrated to be effective in HCC. In 2008, Sorafenib, an anti-angiogenic tyrosine-kinase inhibitor (TKI), was the first drug to show improvement in OS in HCC in the SHARP and AP trials, with an increase in median Overall Survival (OS) from 7.9 to 10.7 months, with no improvement of time to symptomatic progression [4]. This was confirmed in an Asian population in the AP trial [5]. Sorafenib (and some other anti-angiogenics drugs) failed to show benefit in earlier lines of therapies, as adjuvant treatment to either intra-arterial therapies or surgery [6,7,8]. It was almost 10 years before other anti-angiogenic drugs were added, either in second-line after sorafenib (regorafenib, cabozantinib, rivoceranib –also known as apatinib-, all four being TKIs, and ramucirumab, an anti-VEGFR2 antibody), or as non-inferior to sorafenib in first-line (lenvatinib a TKI) [9,10,11,12]. After initial promising results in single-arm studies, the single-agent immune checkpoint inhibitors (ICI) targeting PD-1, nivolumab and pembrolizumab, failed to meet their pre-defined endpoints in randomized phase III trials, respectively, in first-line against sorafenib or in second-line against placebo [13,14]. However, these trials confirmed the activity of the drugs, and paved the way for a combination based on anti-PD-(L)1 targeting to be compared with sorafenib.

To improve results over anti-PD-(L)1 monotherapy, two combination strategies were tested: inhibiting the VEGF/VEGFR pathway, and inhibiting the CTLA-4 pathway. The inhibition of the VEGF/VEGFR pathway can be achieved mostly with two classes of agents: monoclonal antibodies such as bevacizumab (targeting VEGF) or ramucirumab (targeting VEGFR2), or through TKIs, which usually target different tyrosine kinase in complement to VEGFR2 (this class includes sorafenib, lenvatinib, cabozantinib, regorafenib, and rivoceranib). The first approach might provide synergistic results, as the VEGF pathway is also associated with the regulation of immune cells involved in the negative regulation of anti-tumor immunity, such as regulatory T (Treg) cells, myeloid-derived suppressor cells (MDSCs), and tumor-associated macrophages (TAMs) with pro-tumor phenotypes [15]. The second approach targets another step of the cancer-immunity circle, the priming and activation of cytotoxic T cells [16]. Both combination strategies were recently shown to improve survival when compared to sorafenib in first-line treatment of advanced HCC.

## 2. New Standard of Care and Possible Challengers

### 2.1. Results of the Atezolizumab-Bevacizumab Combination Have Been Disruptive

After more than 10 years without change in the first-line standard of care for systemic therapy, progress was long overdue, and IMbrave150 finally delivered a positive result. This trial randomized 501 patients between an atezolizumab (an anti-PD-L1 antibody) plus bevacizumab (an anti-VEGF antibody) combination, and sorafenib. Indeed, the improvement of median OS by almost 6 months with the atezolizumab-bevacizumab combination (from 13.4 [95%Confidence Interval (CI): 11.4–16.9] to 19.2 [95%CI: 17.0–23.7] months), with a Hazard Ratio of 0.66 [95%CI: 0.52–0.85]) was a major leap [17,18]. However, results of IMbrave150 have also been disruptive due to clear evidence of a clinically-meaningful benefit from secondary endpoints. The improvement of objective response rate (ORR) to 30% [95%CI: 25–35%] per Response Criteria In Solid Tumor version 1.1 (RECIST 1.1) opens the way to downstaging strategies, as well as neoadjuvant approaches. Also, and maybe more significantly, the adverse event profile was clearly improved with the new combination, with a lower incidence of disturbing adverse events (AEs) such as hand-foot skin toxicity, fatigue, and diarrhea. Additionally, there was an improvement in time to clinically-significant deterioration of quality of life (QoL), with a median of 11.2 [95%CI: 6.0–Not Reached] vs. 3.6 [95%CI: 3.0–7.0] months for sorafenib [17,19]; the analysis of patients reported outcomes confirmed this combination of efficacy and good safety profile, showing a clinically meaningful improvement in patient-reported QoL [19].

The results were largely confirmed in a phase III trial of a similar combination, the sintilimab (an anti-PD-1 antibody)—IBI305 (a biosimilar of bevacizumab), validated in the ORIENT-32 phase 2/3 trial vs. sorafenib [20]. This trial enrolled 595 patients in China, and showed similar gain in OS (median not reached vs. 10.4 months [95%CI 8.5–NR], HR = 0.57 [95%CI 0.43–0.75], *p* < 0.0001), PFS (median 4.6 [95%CI 4.1–5.7] vs. 2.8 months [2.7–3.2], HR = 0.56 [95%CI 0.46–0.70], *p* < 0.0001) and ORR (21% vs. 4% according to RECIST, *p* < 0.0001) as the atezolizumab-bevacizumab combination. As this trial did not included patients outside China, approval was not sought in Western countries.

These results clearly contrast with previous gains obtained with single-agent antiangiogenics, associated with a more modest gain of median OS (around 3 months), very modest response rates, and no improvement or even deterioration of QoL [4,9,10,11]. Lenvatinib demonstrated only non-inferiority with sorafenib, but was also associated with some improvement in the secondary endpoints, ORR and Progression-Free Survival (PFS) [12].

### 2.2. Two Different Types of Combination Are Currently Tested in First-Line Versus Sorafenib or Lenvatinib

#### 2.2.1. Immunotherapy Combined with an Anti-Angiogenic Tyrosine-Kinase Inhibitor (TKI)

The rationale for these combinations is straightforward and was based on combining a TKI with demonstrated activity in HCC with an anti-PD-(L)1 antibody. Inhibition of the VEGF(R) pathway of the TKI would provide similar synergistic effects as bevacizumab; however, the inhibition of other tyrosine kinase might also improve the anti-tumoral effect. Different combinations have been tested in phase 1b trials, but we will focus on the results of combinations that were provided in phase 3 trials.

The cabozantinib-nivolumab combination showed an ORR of 17% (*n* = 36) in a phase 1 trial, while the triplet cabozantinib-nivolumab-ipilimumab showed a 26% response rate (*n* = 35); median PFS were 5.5 and 6.8 months, respectively [21]. However, the triplet was associated with a higher rate of discontinuation due to AEs (20%) vs. the doublet (3%). The dose of cabozantinib was decreased to 40mg in the phase 3 trial of combination with atezolizumab (which is also consistent with 62% of patients that required dose reduction in the CELESTIAL trial of cabozantinib monotherapy and with the median average daily dose of 35.8 mg) [10]. Results of the phase 3 trial COSMIC-312 were recently published [22]. The trial had 3 arms, sorafenib as the control arm (*n =* 217), cabozantinib-atezolizumab as the experimental arm (*n* = 432), and a cabozantinib alone arm (*n* = 188) to be able to differentiate the differential effect of the combination over cabozantinib alone, even if no formal statistical comparison was planned. The trial had two co-primary endpoints, PFS and OS. The PFS endpoint was met: median PFS was 6.8 months (99%CI 5.6–8.3) in the combination treatment group versus 4.2 months (2.8–7.0) in the sorafenib group (hazard ratio [HR] 0.63, 99%CI 0.44–0.91, *p* = 0.0012); however, at the time of this interim analysis, there was no suggestion of clinically-meaningful benefit in terms of OS: median OS was 15.4 months (96%CI 13.7–17.7) in the combination treatment group versus 15·5 months (12.1–not estimable) in the sorafenib group (HR 0.90, 96%CI 0.69–1.18; *p* = 0.44). ORR was quite low at 13% in the cabozantinib-atezolizumab arm vs. 5% in the sorafenib arm. Grade 3–4 AEs were more frequent in the combination arm (64%) vs. the sorafenib arm (46%).

The combination of lenvatinib-pembrolizumab was tested in large phase 1b clinical trials (*n* = 104). The doses used were the same as monotherapies. Confirmed ORR were 36% per RECIST 1.1 and were associated with a median duration of response of 12.6 months [95%CI: 6.9-not reached]. The median OS was 22 months. However, grade 3-and-more treatment-related AEs were observed in 67%, with 3% of treatment-related deaths [23]. The results of the LEAP-002 phase 3 trial were recently presented at ESMO 2022 [24]. The study compared lenvatinib to the lenvatinib-pembrolizumab combination, and 794 patients were randomized. The median OS with lenvatinib-pembrolizumab was 21.2 months vs. 19.0 months with lenvatinib, and the HR was 0.840 (95%CI: 0.708–0.997, *p* = 0.0227); however, this did not reach the prespecified boundaries (*p* < 0.0185). The HR for PFS was 0.867 (95%CI: 0.734–1.024, *p* = 0.0466), again not crossing the prespecified boundaries (*p* < 0.002). ORR was 26.1% for lenvatinib-pembrolizumab vs. 17.5% for lenvatinib. Grade 3–5 treatment-related adverse events (TRAEs) were 62.5% in the lenvatinib-pembrolizumab arm and 57.5% in the lenvatinib arm (grade 5 TRAEs, 1.0% vs. 0.8%). No QoL data have been presented yet.

Finally, the camrelizumab (an anti-PD-1) and rivoceranib (formely known as apatinib) combination was tested in a phase 2 trial in first-line (*n* = 70) or second-line (*n* = 120), in a Chinese population [25]. Apatinib was previously demonstrated to be an efficacious second-line TKI in Chinese patients, and was approved for this indication in China. For the combination with camrelizumab, ORR was 34.3% in first-line and 22.5% in second-line, with respective median PFS of 5.5 and 5.7 months, and 12-month survival rates of 74.7% and 68.2%. Grade 3-or-more treatment-related AEs were reported in 77.4%, with 1% treatment-related death. The results of the phase 3 trial were also presented at ESMO 2022 [26]. The trial randomized 543 patients between the camrelizumab-rivoceranib combination and a sorafenib control arm. The population was comprised only of 17.3% patients outside Asia, with 74.5% with HBV-related HCC. Both primary endpoints of OS and PFS were met, with a median OS of 22.1 months vs. 15.2 months, HR of 0.62 (95%CI: 0.49–0.80, *p* < 0.001), and median PFS of 5.6 vs. 3.7 months, HR of 0.52 (95%CI: 0.41–0.65, *p* < 0.001). ORR were 35.2% vs. 8.9%. Grade 3/4 AEs were 80.5% vs. 52%, with only 0.4% treatment-related death. Main AEs were hypertension and proteinuria, but also palmar-plantar erythrodysethesia and diarrhea.

Overall, none of these combinations are likely to challenge the atezolizumab-bevacizumab combination as the first-line treatment of choice. Despite meeting the PFS endpoint, the COSMIC-312 did not suggest any benefit in terms of OS [22]. One potential explanation is the increased toxicity of the combination, which required a decreased starting dose of cabozantinib, and, despite this initial dose reduction, 79% of patients required dose interruptions for cabozantinib, and 60% required a further decrease in dose. These dose reductions might have led to lower VEGFR inhibition, resulting in less synergy of the combination. In the LEAP-002 trial, the lenvatinib control arm performed better than anticipated, which, combined with the stringent statistical design, might have resulted in the negativity of the trial despite efficacy of the combination. The double-blind design of the LEAP-002 might also have influenced the investigators to continue lenvatinib, while investigators of the open-label trials with sorafenib as a control arm might have favored earlier discontinuation of the drug in case of toxicities or early signs of progression. Also, the superior PFS for lenvatinib vs. sorafenib demonstrated in the REFLECT might have made it more difficult to show a difference with the combination as compared to other trials using sorafenib. Subsequent therapies, including the use of immunotherapy, might also have contributed to diluting the effect of the combination on OS in both COSMIC-312 and LEAP-002; while this might also have been the case in the IMbrave150 trial, patients in the experimental arm had, at progression, access to TKIs, while, in the other trials, the efficacy of TKIs after cabozantinib or lenvatinib remains unclear. However, subsequent immunotherapy was also more frequent in the camrelizumab-rivoceranib trial than in the LEAP-002 trial, suggesting that this might not be the more important factor contributing to lack of OS benefit. Indeed, the increased toxicity of TKIs over bevacizumab might have made it difficult to challenge the well-tolerated atezolizumab-bevacizumab combination, even if the trials have been clearly positive. Finally, the impact of multi-target TKIs on the tumor microenvironment is not well-documented, and there is a possibility that inhibition of kinases beyond VEGR might have a negative impact on the immune system. Rivoceranib, being more specific to VEGFR, might have led to better synergy as compared to broad-spectrum TKIs such as cabozantinib and lenvatinib. As regards the camrelizumab-rivoceranib combination, the low proportion of non-Asian participants might it make it difficult to gain approval, even if there is no sign of difference of activity in this subgroup. More importantly, even if the results mirror the improved efficacy of atezolizumab-bevacizumab, the toxicity profile seems less favorable, and no QoL data have been presented.

#### 2.2.2. Immunotherapy Combinations with an Anti-CTLA-4 Antibody

The results of tremelimumab used in monotherapy have been published previously, with some evidence of activity in a small phase 2 trial focusing on the Hepatitis-C Virus population (*n* = 21): ORR was 17%, and median OS in second-line was 8.2 months [27]. However, initial results of anti-CTLA-4 combinations with anti-PD-(L)1 were promising. The nivolumab-ipilimumab arm of the Checkmate-040 study compared 3 regimen of the combination: nivolumab 1 mg/kg plus ipilimumab 3 mg/kg, administered every 3 weeks (4 doses), followed by nivolumab 240 mg every 2 weeks (arm A); nivolumab 3 mg/kg plus ipilimumab 1mg/kg, administered every 3 weeks (4 doses), followed by nivolumab 240 mg every 2 weeks (arm B); nivolumab 3 mg/kg every 2 weeks plus ipilimumab 1 mg/kg every 6 weeks (arm C) [28]. While ORR were not clearly different between the 3 arms (32% [95%CI: 20–47%] in arm A, 27% [95%CI: 15–41%] in arm B, and 29% [95%CI: 17–43%] in arm C), median OS appeared higher in arm A (not reached [8.3–33.7+] in arm A, 15.2 months [4.2–29.9+] in arm B, and 21.7 months [2.8–32.7+] in arm C). However, toxicity was also higher in arm A, notably with more patients discontinuing treatment due to AEs (22% vs. 6% in arm B and 2% in arm C), suggesting that high and repeated doses of ipilimumab were associated with both increased efficacy and toxicity [28]. The results of the Checkmate-9DW phase 3 trial (clinicaltrials.gov identifier NCT04039607) which compares the combination of Ipilimumab and nivolumab with sorafenib or lenvatinib are eagerly awaited.

Study 22 evaluated different regimens of the durvalumab(D)-tremelimumab(T) combination: T300+D with a single dose of tremelimumab at 300 mg (so-called STRIDE regimen) and T75+D with 4 doses of tremelimumab 75 mg every 4 weeks, each of them combined with Durvalumab 1500 mg every 4 weeks [29]. In the study, single-agent arms (durvalumab 1500 mg every 4 weeks and tremelimumab 750 every 4 weeks) were also included. The T300+D arm (*n* = 75) was associated with higher ORR (24% vs. 10% for the T75+D (*n* = 84), 11% for durvalumab (*n* = 104) and 7% for tremelimumab (*n* = 69)), and higher median OS (18.7 months vs. 11.3 for T75+D, 13.6 for durvalumab and 15.1 months for tremelimumab) [29]. Interestingly, this single high dose was not associated with an increase in discontinuation of treatment due to AEs (although it was associated with a higher frequency of grade 3/4 treatment-related AEs as compared with durvalumab). The results of the HIMALAYA phase 3 trial were recently published [30]. A total of 1171 patients were randomized between the STRIDE regimen, durvalumab monotherapy, and sorafenib. The median OS was 16.43 months (95% confidence interval [CI], 14.16 to 19.58) with STRIDE, 16.56 months (95%CI: 14.06 to 19.12) with durvalumab, and 13.77 months (95%CI: 12.25 to 16.13) with sorafenib. The OS hazard ratio for STRIDE versus sorafenib was 0.78 (96.02%CI: 0.65 to 0.93; *p* = 0.0035). Median PFS was not different between the 3 arms. ORR were 20% with STRIDE, 17% with durvalumab, and 5% with sorafenib. Grade 3–4 AEs occurred in 51% of patients receiving STRIDE, 37% of patients receiving durvalumab, and 52% of patients receiving sorafenib. Time to deterioration of QoL was also improved with STRIDE and durvalumab as compared with sorafenib.

The results of the HIMALAYA trial provide the STRIDE regimen as an alternative to the atezolizumab-bevacizumab combination. The results showed clinically-meaningful and statistically-significant improvement of OS, ORR, and QoL. The STRIDE regimen was recently granted approval by the FDA [31].

The results of the discussed phase 3 trials are presented in Table 1. Of note, cross-trial comparisons are always at risk of misinterpretation. Some inclusion and exclusion criteria were different between the trials; there were also differences in the population included, for example, as regards country and underlying liver disease, and these parameters might influence efficacy and safety, as well as subsequent treatment practices. Even newer methodologies, such as matching-adjusted indirect comparison, may show conflicting results when applied to the same comparison [32,33]. For example, there was a very different ORR for the same control arm of sorafenib between IMbrave150 (11%) and HIMALAYA (5%), making cross-trial comparison of the ORR of the experimental arms hazardous.

### 2.3. Potential Criteria of Choice between the Validated Combinations

As we now have different combinations with demonstrated activity, we will be confronted with a choice between different first-line treatment options.

Most clinical trials used broadly similar criteria as regards the population included, focusing on patients with Child-Pugh A liver disease and Performance Status 0–1. A potential differentiation will be for combinations without antiangiogenics to be preferable in patients with cardiovascular comorbidities or significant portal hypertension. In the initial trials of bevacizumab monotherapy in HCC, severe esophageal bleedings were seen in 11% of patients [34]. This was decreased in IMbrave150, probably due to the routine use of pretreatment endoscopy and management of esophageal varices. In contrast, patients with pre-existing auto-immune disease (not requiring systemic steroids, which would be considered contra-indicated for anti-PD-(L)1 antibodies) might be preferred for combination without anti-CTLA-4 antibodies. However, such patients will clearly be a large minority of the population, and most of the patients will not have clinical criteria that are clearly contra-indicating for one combination but not the others. Weaker criteria might also be derived from these strong contra-indications (Table 2):

In the future, research on potential predictive factors of the efficacy of immunotherapy combination would be of paramount importance. While the study of patients treated with anti-PD-1 single-agents has already shown some potential biomarkers of response [35,36,37], it would also be interesting to define predictive factors that might favor the use of antiangiogenics or anti-CTLA-4. Indeed, the biomarker study of the Study-22 suggested that the proportion of proliferative CD8+ cells in the blood was associated with response; also, the expansion of clonality was associated with response and survival, and more clearly so in the CTLA-4-containing arms [29,38]. In renal cell carcinoma, in a phase 2 study, the benefit of adding bevacizumab to atezolizumab seemed limited to the population of patients with both high effector T cells and high myeloid cells in the tumors [39]. The recent analysis of the biomarker program of the development of the atezolizumab-bevacizumab combination suggested interesting potential predictive factors [40]. Pre-existing high expression of PD-L1, T-effector signature and intratumoral CD8+ T cell density were associated with better outcomes. The benefit of the bevacizumab was suggested for tumors with high expression of VEGFR2, Tregs, and myeloid inflammation signatures. These findings might suggest that patients whose tumors harbor such signatures are better candidates for anti-angiogenics-based combination. although further validation is required.

Recently, data suggested that patients with underlying NAFLD might derive less benefit from immunotherapy [41]. Preclinical data were supported by a meta-analysis of some randomized trials that suggested a lower efficacy of immunotherapy in non-viral HCC in terms of OS. However, the recent results of the HIMALAYA, LEAP-002, and camrelizumab-rivoceranib trials did not confirm such lower efficacy in terms of OS, and, in many of the previous trials, efficacy was suggested in non-viral HCC in terms of ORR and PFS, raising doubts about the general interpretation of the initial observations.

### 2.4. Is There Still a Role for Monotherapy in the First-Line Setting?

Anti-PD-1 single agents have been compared to sorafenib as first-line treatment. In the Checkmate-459, nivolumab failed to demonstrate superiority, despite a positive trend in OS (median OS of 16.4 vs. 14.7 months, HR = 0.85 (95%CI: 13.9–18.4, *p* = 0.075 for superiority), a better toxicity profile, and improved QoL. Similar results were recently reproduced in 2 comparisons that, this time, used a non-inferiority design: in the HIMALAYA trial, the single-agent durvalumab arm (median OS of 16.6 vs. 13.8 for sorafenib, HR = 0.86, 95.7%CI: 0.73–1.03, non-inferiority margin of 1.08 not crossed) and in the rationale-301 trial comparing the anti-PD-1 antibody tislelizumab to sorafenib (median OS 15.9 vs. 14.1 months, HR = 0.85 (95%CI: 0.71–1.02, *p* = 0.039 for non-inferiority with a margin of 1.08)). However, due to the superiority of combination therapies in phase 3 trials, it is difficult to see the roles of single-agent anti-PD-(L)1 (excluding the discussion about limitation due to approval and coverage of the combinations in each country). In the case of contra-indication to anti-angiogenics, it is probable that the durvalumab-tremelimumab combination would be preferred, and there is no suggestion that atezolizumab-bevacizumab might cause more immune-induced toxicity as compared to anti-PD-1 monotherapy. Single-agent anti-PD-1 agents might then be tested in populations with increased risk of toxicity, such as patients with Child-Pugh B liver function, Performance Status 2, or the elderly; however, validation of such strategies should be conducted.

Furthermore, while the LEAP-002 study might be viewed as a validation of the efficacy of lenvatinib, the REFLECT study did not show superiority over sorafenib [12]. Due to the worse toxicity profile, it will be difficult to keep a role for first-line lenvatinib as compared to combinations that demonstrated superiority. The subgroup of patients with clear contra-indication to immunotherapy, such as transplant patients or severe uncontrolled pre-existing auto-immune disease patients, will, however, still benefit from the single-agent TKIs sorafenib or lenvatinib.

## 3. Current and Future Second-Line Options in the Area of First-Line Immunotherapy-Based Combination

### 3.1. Available Guidelines for Treatment after Progression on Atezolizumab-Bevacizumab

Previous recommendations for second-line trials were based on phase III trials performed after sorafenib as first-line treatment. There are currently limited data on the efficacy of TKIs following immunotherapy-based combinations. These data suggest similar efficacy and safety as when used in first-line [42,43]. Despite this paucity of data, most updates of guidelines do recommend the use of TKIs after progression on first-line atezolizumab-bevacizumab. However, some important differences appear with regard to the relative sequences recommended after atezolizumab-bevacizumab (Table 3).

### 3.2. What Will Be the Optimal Sequence of Treatment?

With the availability of different combinations validated in first-line, the question of the efficacy of sequential treatment with these different combinations will arise (Figure 1).

As different TKIs have been validated in sequences, similar sequential treatments might be efficacious in the immuno-oncology era. However, this cannot be taken for granted. One such question might be the optimal sequence between anti-CTLA-4 and anti-angiogenics-based combinations. Currently, no data exist to inform this important question. Another question will be the role of continued inhibition of the PD-L1/PD-1 pathway following progression on an anti-PD-(L)1-containing regimen. In melanoma, retrospective data suggest that the combination of nivolumab with ipilimumab is superior to monotherapy ipilimumab after failure on anti-PD(L)1 therapy [45]. A last question will be the differences between the different anti-angiogenics used.

Various arguments can be proposed to justify continuation of anti-PD-(L)1 inhibition after progression on a anti-PD-(L)1 based combination: 1—using a different type of combination (anti-CTLA-4 after antiangiogenics) might counteract the resistance to anti-PD-(L)1 inhibition; 2—continuing anti-PD-(L)1 but using a TKI after bevacizumab might have a differential effect due to the larger spectrum of inhibition of TKIs, non-limited to antiangiogenic effect as compared to bevacizumab; 3—use of antibodies might lead to the development of antidrug antibodies (ADA), for which a functional role in oncology is still unclear but might be associated with decreased efficacy, and which could then justify the switch for a different drug [46,47]. Atezolizumab has been associated with a high rate of ADA development, with data suggesting reduced efficacy in patients who developed ADA [47]. A recent prospective validation of the development of atezolizumab-directed ADA as a negative predictive factor of response has been published [48]. In this study, development of ADA at 3-weeks was found in 17% of patients to associated with decreased response rates (11% vs. 34% in the discovery cohort [*n* = 50], 7% vs. 29% in the validation cohort [*n* = 82]), as well as worse PFS (HR = 2.84, *p* = 0.005 in the discovery cohort, HR = 2.52, *p* = 0.006 in the validation cohort) and worse OS (HR = 3.30, *p* = 0.003 in the discovery cohort, HR = 5.81, *p* = 0.001 in the validation cohort). However, the cost and toxicities associated with strategies of continuation of anti-PD-(L)1 inhibition after progression justify their use only in adequately-designed prospective studies.

One study addressing this question is the IMbrave251 trial, a phase 3 trial (*n* = 554) in which patients progressing after first-line atezolizumab-bevacizumab are randomized to lenvatinib vs. lenvatinib-atezolizumab combination (clinicaltrials.gov identifier NCT04770896). Other emerging questions will be the potential effectiveness of anti-CTLA-4-based combination after anti-angiogenic-based combination, or the reverse sequence.

### 3.3. New Treatment Strategies for Advanced Disease

The development of treatment associated with non-anecdotal objective responses, and with different efficacy/toxicity profiles, will raise new possibilities in the advanced disease. New strategies might now be foreseen (Figure 2).

The traditional aim in the advanced setting is palliative control (Figure 2A); in this context, the choice of treatment will favor an efficient combination with lower toxicity, to preserve QoL as long as possible. However, the ORR seen with recent systemic treatment raises the possibility of redirecting patients for a local treatment of tumors that were initially outside criteria (for example outside transplant criteria, Figure 2B). An intermediate situation is the context of a large tumor burden, or a lesion at risk of local complication (for example, portal vein thrombosis). In the context of portal vein thrombosis of the main portal vein (Vp4), atezolizumab-bevacizumab was suggested to improve results over sorafenib (median OS of 7.6 vs. 5.5 months); however, the low median OS illustrates that the medical need remains incompletely met for this population [49]. In this context, we might question the role of combining locoregional treatment to immunotherapy-based combination to improve results (Figure 2C). With the development of alternative treatment regimens, the clinician might have different preferences depending on the objective: for example, when downsizing is the goal, the choice of a combination with higher response rates might be preferred; when a combination with local treatment, anti-CTLA-4-based combinations might have stronger rationale than antiangiogenics-based combinations, as the antiangiogenics might be at risk for complications, without intrinsic potential synergy.

### 3.4. Development of Immunotherapy-Combination in Earlier Stages, and Potential Difficulties

After the results are demonstrated in the advanced stage, many clinical trials have been launched in earlier stages (Table 4).

## 4. Challenges for Future Development

### 4.1. Building Evidence on Sequencial Treatment

The ideal method of navigating different sequence options would be to perform a randomized trial of different strategies, including successive lines of treatment. However, performing prospective clinical trials to answer questions about sequences might be very difficult due to the high number of patients required to show benefit (and particularly survival benefit) between two sequences. Moreover, investigators and patients might feel reluctant to perform trials if they have access to combinations validated in the first-line setting. Finally, any benefit of each line might be diluted by the efficacy of further lines, making the results particularly difficult to interpret. There is, thus, a high probability that most data will come from retrospective analyses. The inherent biases of such analyses might make it difficult to identify the optimal sequences.

### 4.2. Backbone for New Combinations

Despite this revolution of immunotherapy with significant increase in response rates, the majority of patients still do not show objective responses, and many patients with stable disease experience short control of the disease. There is still an urgent need to improve systemic therapy which will require new strategies. Adding new agents, mainly with immune-related mechanisms of action, is one of the obvious avenues. With the development of various novel immune checkpoint inhibitors (targeting, for example, TIGIT, LAG3, etc.), questions are already arising regarding further combinations. One first question might be the triplet combination targeting PD-(L)1, CTLA-4, and VEGF. With different first-line regimens validated, this question will be the best backbone to develop a triplet combination. Frequently, the available portfolio of each pharma company is the main criterion for selection of the backbone. However, questioning the rationale for triplet combination might be of paramount importance, as we will not be able to thoroughly test all potential combinations. Building virtual clinical trials might be a method for selecting a promising combination, as was recently done for HCC with the anti-PD-1/anti-CTLA-4 combination [50]. Other predictive models have also been developed [51]. Another key factor will also be the risk of overlapping toxicities, which will once again favor the less toxic combinations. Some predictive models of toxicities might help to select combinations for clinical testing, although predicting toxicity to ICI might be more difficult [52]. Finally, cost will be of paramount importance from a global point of view, as most cases of HCC arise in low-to-middle-income countries, adding more pressure to find predictive factors of efficacy when compared to sorafenib.

### 4.3. Predictive Biomarkers of Response for Personalized Therapy

As previously discussed, the rational development of predictive biomarkers of response to new agents are of major importance. Contrary to other tumors, and despite some limited initial signals, PD-L1 expression is not sufficiently discriminative, and Tumor Mutational Burden is not frequently high in HCC [53]. Preliminary data on the atezolizumab +/− bevacizumab trials, similar to the durvalumab +/− tremelimumab trials, suggest that some parameters might influence response, although, currently, no validated biomarkers exist. During drug development, emphasis should be given to appropriate biobanking, especially in the HCC context in which tumor biopsy is still considered by some clinicians as not required in the routine care.

### 4.4. Financial Burden

As in many fields in oncology, the development of new agents is associated with increased cost for the society, which should be compensated for by improved benefit to the patients and society. Limited data are available as to cost-effectiveness of immunotherapy in HCC, and results, by nature, will depend on the context of each health-care system. For example, an analysis conducted from a Chinese perspective suggested that the sintilimab-bevacizumab biosimilar combination might be cost-effective depending on the cost of sintilimab, while the atezolizumab-bevacizumab combination would not be [54]. Similarly, analyses of atezolizumab-bevacizumab from a US-payer perspective suggested that a substantial decrease in the cost of atezolizumab would be needed to achieve cost-effectiveness [55,56].

## 5. Conclusions

The development of immunotherapy-based combinations has already changed the field of advanced HCC. Apart from changing first-line treatment, many questions will arise in the near future. Potential various alternative first-line options from the atezolizumab-bevacizumab combination have already arisen with the clear positivity of the durvalumab-tremelimumab phase 3 trial, as well as the camrelizumab-rivoceranib phase 3 trial. The development of new strategies, including downstaging strategies and/or combination with locoregional treatment, might help clinicians to make the choice between alternative first-line options. Future combinations will be built upon existing ones, and the choice of the appropriate backbone should, ideally, be based on scientific rationale rather than industrial opportunity. The definition of the best sequencing of treatments will require academic efforts. The development of predictive biomarkers to guide which type of combination is required for each patient would be a major leap in the direction of personalized therapy for HCC. The fact that so many questions are now raised in regard to advanced HCC is clearly an indicator of the changes induced by the immunotherapy-based combinations. While atezolizumab-bevacizumab combination is a major advance, it might just be the first step of a revolution.

## Figures and Tables

**Figure 1 cancers-14-05868-f001:**
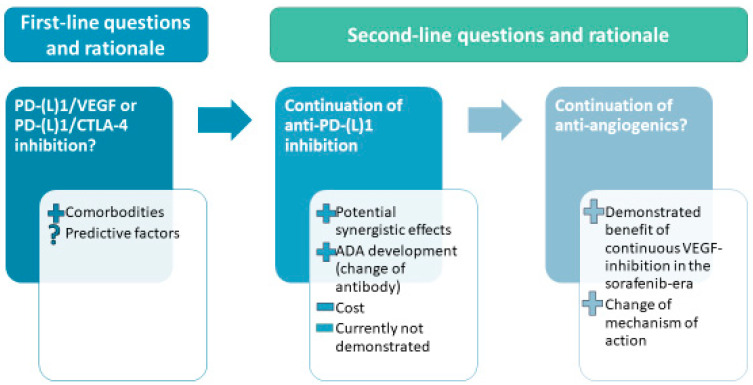
Different questions raised by the possibility of sequences, and rationale behind these questions. ADA: antidrug antibodies.

**Figure 2 cancers-14-05868-f002:**
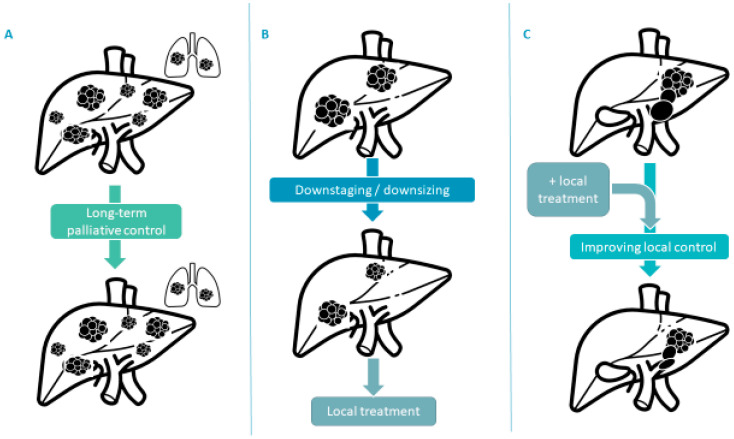
Potential emerging strategies for use of systemic therapies in unresectable HCC: (**A**)—long-term palliative control, (**B**)—downstaging to local treatment (resection, transplantation, percutaneous destruction, transarterial treatment, …), and (**C**)—combination with local treatment for improved local control.

**Table 1 cancers-14-05868-t001:** Main results of the presented phase III trials of immunotherapy-based combination compared to sorafenib as first-line treatment of hepatocellular carcinoma. OS: Overall Survival; PFS: Progression-Free Survival; ORR (Objective Response Rate); MVI: Macrovascular Invasion; HR: Hazard Ratio; * not statistically-significant per statistical assumptions; Vp4: Tumoral Portal Vein Thrombosis reaching the main portal vein.

Combination	Control Arm	Trial Name and Reference	Selected Characteristics Population Included	*n* Main Comparison	OS Experimental vs. Control	PFS Experimental vs. Control	RECIST 1.1 ORR Experimental vs. Control	Treatment-Emergent Grade 3/4 Toxicities Experimental vs. Control
Atezolizumab-bevacizumab	sorafenib	IMbrave150 [17,18]	MVI: 40% (14% Vp4) Non-viral: 30.5%	501	Median 19.2 vs. 13.4 months, HR = 0.66, *p* < 0.001	Median 6.8 vs. 4.3 months, HR = 0.29, *p* < 0.001	30% vs. 11%, *p* < 0.001	63% vs. 57%
Lenvatinib-pembrolizumab	lenvatinib	LEAP-002 [24]	MVI: 17% (0% Vp4) Non-viral: 39%	794	Median 21.2 vs. 19.0, HR = 0.840, *p* = 0.0227 *	Median 8.2 vs. 8.0 months, HR = 0.87, *p* = 0.0466 *	26% vs. 18%	62% vs. 57%
Camrelizumab-rivoceranib	sorafenib	Qin et al. [26]	MVI: 17% (0% Vp4) Non-viral: 19.2%	543	Median 22.1 vs. 15.2 months, HR = 0.62, *p* < 0.001	Median 5.6 vs. 3.7 months, HR = 0.52 *p* < 0.001	35.2% vs. 8.9%	81% vs. 52%
Cabozantinib-atezolizumab	sorafenib	COSMIC-312 [22]	MVI: 30% (18% Vp4) Non-viral: 39%	649	Median 15.4 vs. 15.5 months, HR = 0.90, *p* = 0.44	Median 6.8 vs. 4.2 months, HR = 0.63, *p* = 0.0012	11% vs. 4%	64% vs. 46%
Durvalumab-tremelimumab	sorafenib	HIMALAYA [30]	MVI: 26% (0%Vp4) Non-viral: 42%	782	Median 16.4 vs. 13.8 months, HR = 0.78, *p* = 0.0035	Median 3.8 vs. 4.1 months, HR = 0.90	20% vs. 5%	51% vs. 52%

**Table 2 cancers-14-05868-t002:** Potential criteria for the choice of future first-line treatment.

	Anti-Angiogenics-Based		Anti-CTLA-4-Based
	Bevacizumab-Based	TKIs-Based	
Strong criteria of choice	Pre-existing untreated auto-immune disorder (psoriasis…)		Severe vascular comorbidities Significant Portal Hypertension
Criteria of choice subject to interpretation	Better toxicity profile		Non-severe vascular comorbidities Equilibrated arterial hypertension
	Low-risk auto-immune disorder (diabetes, thyroid disorder…)		

**Table 3 cancers-14-05868-t003:** Existing guidelines regarding subsequent treatment after atezolizumab-bevacizumab combination.

	Second-Line Options	Further-Line Options
BCLC [3]	Clinical trials	
EASL [44]	Multi-TKI and VEGFR2 inhibitor as per off-label availability	
ILCA, https://ilca-online.org/education/ilca-guidances/ (accessed on 24 November 2022)	sorafenib, lenvatinib, cabozantinib	regorafenib
NCCN preferred treatments	sorafenib, lenvatinib, regorafenib, cabozantinib, ramucirumab	
ESMO https://www.esmo.org/guidelines/gastrointestinal-cancers/hepatocellular-carcinoma (accessed on 24 November 2022)	sorafenib, lenvatinib, cabozantinib, ramucirumab	regorafenib (for TKI-experienced patients)
French recommendation (TNCD)	sorafenib, lenvatinib	regorafenib, cabozantinib, ramucirumab

**Table 4 cancers-14-05868-t004:** Ongoing randomized trials testing immunotherapy in earlier stages of HCC. TACE: Trans-arterial Chemoembolization; SIRT: Selective Internal Radiation Therapy.

Trial Name	Context	Treatment	Planned Number of Patients	Clinicaltrials.gov Identifier
IMbrave 050	After curative-intent resection or ablation	Atezolizumab-bevacizumab vs. surveillance	668	NCT04102098
EMERALD-2	After curative-intent resection or ablation	Durvalumab +/− bevacizumab vs. placebo	908	NCT03847428
Checkmate-9DX	After curative-intent resection or ablation	Nivolumab vs. placebo	545	NCT03383458
Keynote-937	After curative-intent resection or ablation	Pembrolizumab vs. placebo	950	NCT03867084
ABLATE-2	Before and after ablation	Atezolizumab-bevacizumab + ablation vs. ablation alone	202	NCT04727307
ML42612	With TACE	Atezolizumab-bevacizumab + TACE vs. TACE alone	342	NCT04712643
EMERALD-1	With TACE	Durvalumab +/− bevacizumab + TACE vs. placebo + TACE	724	NCT03778957
EMERALD-3	With TACE	Durvalumab-tremelimumab +/− lenvatinib + TACE vs. TACE alone	525	NCT05301842
TACE-3	Before and after TACE	nivolumab + TACE vs. TACE alone	522	NCT04268888
LEAP-012	With TACE	pembrolizumab-lenvatinib + TACE vs. placebo + TACE	950	NCT04246177
ROWAN	With SIRT	durvalumab-tremelimumab vs. SIRT alone	150	NCT05063565
ABC-HCC	In lieu of TACE	atezolizumab-bevacizumab vs. TACE	434	NCT04803994
INTRATACE	In lieu of TACE	regorafenib-tislelizumab vs. TACE	496	NCT04777851

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
