# Peer review of "New Challenges Facing Systemic Therapies of Advanced HCC in the Era of Different First-Line Immunotherapy-Based Combinations"

_cancers, 2022, doi:10.3390/cancers14235868_

Round 1

Reviewer 1 Report

The article, New challenges facing systemic therapies of advanced HCC in 2 the era of different first-line immunotherapy-based combinations by Edeline et al., outlined the recent advancements in the field of hepatocellular carcinoma. I am quite impressed with the work, especially the way various recent trials such as LEAP-002, COSMIC, IMBrave, and HIMALAYA trials were described and dissected. I congratulate the authors for their excellent work and I believe it would be a good addition to the field and appeal to the readership of Cancers.

Reviewer 2 Report

New challenges facing systemic therapies of advanced HCC in 2 the era of different first-line immunotherapy-based combinations
It is a well-written comprehensive review paper that discusses the treatment options and challenges for advanced hepatocellular carcinoma.

1. Tremelimumab in combination with durvalumab was approved by FDA last month (October, 2022), this must be modified throughout the paper.
(https://www.fda.gov/drugs/resources-information-approved-drugs/fda-approvestremelimumab-combination-durvalumab-unresectable-hepatocellular-carcinoma)
2. Please rewrite Page 3, lines 84-91. It is not clear when is the comparison made to just sorafenib and when to atezo and bevacizumab combination
3. For few cancers type (e.g. NSCLC) TMB is considered an important biomarker. Please mention significance of TMB as biomarker for HCC. Some ref: PMID: 34595140; DOI: 10.1080/13543784.2021.1972969
4. Quantitative systems pharmacology models (QSP) are used to design new combination of drugs. Authors should mention the recently published work on first ever QSP model for HCC that studies combination of immunotherapy (DOI: 10.1136/jitc-2022-005414 ). This work can change the criteria for selection backbone, treatment sequence.
5. Authors should also specify the mechanistic models such as DILIsym that could be used to identify the adverse events and assist in designing of clinical trials. Ref: https://doi.org/10.1016/j.cotox.2020.06.003
6. In table 1, please add a column with name of identifier of the clinical trial to improve the readability. Also mention vp4 in table caption.
7. To overcome the challenges predictive Computational models such as PMID: 30651095; PMID: 34774999; DOI: 10.1109/BSB.2018.8770553 can be used to design personalized therapy for HCC. This could help is designing clinical trials with better efficacy for hcc patients.
8. Minor comment: Since this study compare different anti-angiogenic drugs in combination with other drugs, it would be worth including a section on the difference in the mechanism of action of these different anti-angiogenic drugs such as beva, cabo, lenva etc

Overall, I recommend accepting this paper after revision based on the comments mentioned above.
